# Effects of *Pieris japonica* (Ericaceae) dominance on cool temperate forest altered-understory environments and soil microbiomes in Southern Japan

**Yuji Tokumoto**[1]*, **Ayumi Katayama**[2]

**1** Institute for Tenure Track Promotion, University of Miyazaki, Miyazaki, Japan, **2** Shiiba Research Forest, Kyushu University, Shiiba, Miyazaki, Japan

* tokumoto.ug@gmail.com

**Data Availability Statement:** All the raw sequence data of the soil prokaryotic 16S rDNA, the soil eukaryotic 18S rRNA genes, and the root-associated fungal 18S rRNA genes were submitted in the Sequence Read Archive of DDBJ database

## Abstract

The number of plants unpalatable to deer increases with increasing deer numbers. In the Kyushu Mountain area of Southern Japan, *Pieris japonica* (Ericaceae), an unpalatable shrub, has become the monodominant vegetation under evergreen conifer and deciduous broad-leaved tree stands. The monodominance of unpalatable plants in the understory has potential advantages and drawbacks; however, the effects of *Pieris* dominance are not well understood. To assess the effects of *P. japonica* dominances on forest environments and ecosystems, we investigated understory environments and soil microbiomes in *Pieris*-dominant sites. Under the deciduous broad-leaved trees, *Pieris* dominance leads to considerable *Pieris* leaf litter and humus weights and low soil bulk density and canopy openness. In the soil fungal community and fungal functional groups, the relative abundance of symbiotrophic fungi, particularly ectomycorrhizal fungi in *Pieris*-dominant sites were lower than in other-vegetation understory sites and saprotrophic fungi *vice versa*. Because few seedlings and saplings were found under *Pieris* shrubs, *Pieris* dominance in the understory might exclude other plant species. The results of this study will contribute to the *Pieris* population and forest management following deer overgrazing.

## Introduction

Increasing deer populations have altered forest environments worldwide [1]. Deer overgrazing affects plant covers, which leads to changes in species composition, biodiversity, and environments. Deer overgrazing increases the populations of plants unpalatable to deer. This can lead to unpalatable plants overlaying forest understories, which are referred to as understory canopies [2, 3]. Unpalatable plants have specific physical traits, such as tough leaves and needles, and they contain poisonous chemicals, which prevent deer eating and/or digesting them easily [4]. Although some of these plants can be used for slope revegetation to prevent soil erosion and decrease plant species diversity [5], the effects of increasing unpalatable plants on ecosystems, both positive and negative, are not well understood. In addition, little is known about

under the accession number DRA016984, DRA016985, and DRA016986, respectively. All relevant environmental data are within the manuscript and its Supporting Information files.

**Funding:** Yuji Tokumoto, the Botanical Research Grant of ICHIMURA Foundation for New Technology (No. 2022-15 and 2023-20) Yuji Tokumoto, Leading Initiative for Excellent Young Researchers (LEADER) from the Ministry of Education, Culture, Sports, Science and Technology (MEXT), Japan (Grant Number JPMXS0320220123) Ayumi Katayama, Japan Society for the Promotion of Science (JSPS) Grant-in-Aid for Scientific Research (B) (Grant Number 22H03793). The funders had no role in study design, data collection and analysis, decision to publish, or preparation of the manuscript.

**Competing interests:** The authors have declared that no competing interests exist.

the management of overabundant unpalatable plants compared with the effects of deer overgrazing.

Ericaceae plants include several types of trees unpalatable to deer, and they form the mono-dominant understory layers in temperate and boreal forests in Europe and North America [2]. The effects of Ericaceae dominance have been confirmed by the increased thickness of the humus layers, and changes in soil chemical properties [6, 7]. Ericaceae plants produce poly-phenol-rich litter that contain high levels of tannin, which influence the regeneration of over-story trees, soil organisms, and carbon (C) and nitrogen (N) cycles [6–8]. Ericaceae plants have mutualistic relationships with ericoid mycorrhizae (ErM), and the dominance of the Ericaceae plants can affect other symbiotic fungi, ectomycorrhiza (ECM), and arbuscular mycorrhizae (AM) fungi [7]. The growth of trees from other mycorrhiza types is inhibited and the ECM fungal root colonization is affected by the presence of Ericaceae plants and ErM [9, 10]. Tree species in forest upper layers often affect microbial community compositions, and several tree species-specific fungi have been confirmed in two main mycorrhiza types (i.e., ECM and AM) [11]. However, studies on ErM and ericoid plants are scarce [7, 12, 13], and little is known about Ericaceae plant dominance related to changes in soil microbial communities and plant–soil feedback.

One Ericaceae plant, *Pieris japonica* (Thunb.) D.Don ex G.Don is an evergreen shrub distributed broadly in forest understories and on the open ridges in nutrient-poor soils throughout the Japanese islands of Honshu, Shikoku, and Kyushu [14]. *Pieris japonica* has mutualistic relationships with ErM and AM [15], and is also known to be poisonous to mammals, as it contains asebotoxin and grayanotoxin in the plant organs [14]. In the past, few tree individuals were observed in the matured natural forest and they did not dominate understories. However, after sika deer (*Cervus nippon*) grazing following light environmental changes in the understories, the species population has gradually increased in forest understories and formed understory canopies because sika deer prefer not to eat this plant [14, 16–18]. It has been reported that *Pieris* domination leads to low soil water content and a high decomposition rate of the leaf litter in the short research term (i.e., 3–4 months) [19]; however, other environmental parameters, such as soil properties, microbiomes, and ecological functions, have not been studied. Since these phenomena caused by Ericaceae dominance can also occur in *Pieris*-dominant forests, the effects of expansion of this shrub should be investigated for the purpose of forest management after deer overgrazing.

To investigate *Pieris* dominance in the forest understory, we assessed differences in the understory environments, such as the light environment, chemical and physical soil properties, and microbial communities, between *Pieris*-dominant sites and those *Pieris*-nondominant sites.

## Materials and methods

### Study site

Our study was conducted at the Shiiba Research Forest of the Department of Agro-environmental Sciences, Kyushu University, Shiiba, Miyazaki Prefecture, Japan (32˚23'N, 131˚10'E; 1029–1079 m a.s.l.) located in the southern Kyushu Mountain area. The mean annual precipitation is 3,207.9 mm, and annual mean temperature is 10.8˚C [20]. The monthly mean temperature ranges from −0.3˚C in January to 24.2˚C in August. The natural vegetation is a cool temperate forest comprising deciduous broad-leaved trees—*Fagus crenata* (Fagaceae), *Quercus crispula* (Fagaceae), and *Carpinus* spp. (Betulaceae), evergreen broad-leaved trees—*Quercus salicina* and *Quercus acuta* (Fagaceae), and conifers—*Tsuga sieboldii* (Pinaceae), *Abies firma* (Pinaceae), and *Pinus densiflora* (Pinaceae). Shika deer were frequently observed from 1976–

**Table 1. Research site descriptions with region names, types of canopy trees, understory layers, and numbers of plots per site.**

| Regions | Sites | Types of canopy trees | Tree Genera of canopy trees | Mycorrhizae types of canopy trees | Understory | No. plot |
|---------|-------|----------------------|-----------------------------|-----------------------------------|-----------|----------|
| Maruju | M01 | Coniferous trees | *Pinus* | ECM | *Pieris* | 3 |
| | M02 | Coniferous trees | *Pinus, Abies* | ECM | Other | 3 |
| | M03 | Deciduous broad-leaved trees | *Betula (planted)* | ECM | *Pieris* | 3 |
| | M04 | Deciduous broad-leaved trees | *Carpinus, Quercus* | ECM | Other | 3 |
| | M05 | Coniferous trees | *Abies* | ECM | *Pieris* | 2 |
| | M06 | Coniferous trees | *Abies* | ECM | Other | 2 |
| Hirono | H01 | Coniferous trees | *Abies* | ECM | *Pieris* | 3 |
| | H02 | Coniferous trees | *Abies, Pinus* | ECM | Other | 3 |
| | H03 | Coniferous trees | *Pinus* | ECM | *Pieris* | 3 |
| | H04 | Coniferous trees | *Pinus* | ECM | Other | 3 |
| | H05 | Coniferous trees / Deciduous broad-leaved trees | *Abies, Carpinus* | ECM | *Pieris* | 2 |
| | H06 | Coniferous trees | *Pinus* | ECM | Other | 2 |
| | H07 | Deciduous broad-leaved trees | *Quercus* | ECM | *Pieris* | 2 |
| | H08 | Deciduous broad-leaved trees | *Quercus* | ECM | Other | 2 |

1984 and the population density from 2005–2008 was approximately 20–50 individuals $km^{-2}$ [21]. Because of shika deer overgrazing, the understory vegetation, previously represented by dwarf bamboo (*Sasa borealis*), is now dominated by unpalatable shrubs, such as *P. japonica* (Ericaceae) and *Illicium anisatum* (Schisandraceae) [17]. We selected two study regions, Maruju (division No. 35) and Hirono (division Nos. 22, 23, and 24), where *Pieris* shrubs have been observed in the understory (S1 Fig, Table 1). The soil group of study site is andosole according to the classification system of the IUSS Working Group World Reference Base (2014). In the Maruju region, we selected three *Pieris*-dominant sites and three without *Pieris* (i.e., they have been understory grazed; however, they contain some grasses or mosses and no *Pieris* shrubs have been established) near the *Pieris*-dominant sites. In total, six sites were selected, and 2–3 plots (20 × 20 cm each) were selected for each site. In the Hirono region, we selected eight sites (four Pieris-dominant sites and four without *Pieris*) and selected 2–3 plots per site. For each plot, the types of canopy trees nearby were recorded in addition to detailed site information, including the canopy trees, tree genera, understory layers, and the numbers of plots per site, which are presented in Table 1. We categorized combinations of canopy tree types and understory trees as follows: C-Pieris; sites with a conifer canopy and *Pieris* in the understory, C-Other; conifers and other understory vegetation, such as grasses and mosses, DB-Pieris; deciduous broad-leaved trees and *Pieris* in the understory, and DB-Other; deciduous broad-leaved trees and other understory vegetation. This study was approved by the Committee for Forest Management, Kyushu University Forest, Kyushu University, Japan (#M2023-002).

## Environmental properties of the forest understories

To investigate differences in the environmental properties of the *Pieris*-dominant and other understory sites, we analyzed the light environment and chemical and physical soil properties (Table 2). To analyze the light environment at each plot, fish-eye images of the plots were captured using a 360˚ camera (theta-V; Ricoh, Japan) in June, August, and October 2022, and canopy openness as a percentage was quantified using CanopOn2 software (http://takenaka-akio. org/etc/canopon2/). In June, plot soil hardness was measured four times using a soil hardness meter (Yamanaka, Fujiwara, Japan), and the measurements were averaged per plot. To quantify the litter and humus layers in each plot, we collected $A_0$ horizons using a 20 × 20 cm frame. The litter was air dried and sieved using 4-mm mesh. Particles >4 mm were categorized

**Table 2. Environmental properties of the sampling points with the results of linear-mixed-model analyses of the effects of understory vegetation and canopy trees.**

| Canopy & understory trees | Conifers | | Deciduous broad-leaved trees | | p-values of fixed effects | | |
|---|---|---|---|---|---|---|---|
| Variables | Pieris | Other | Pieris | Other | Understory | Canopy | Interaction |
| Canopy openness (%) June | 41.54 ± 1.18 a | 43.18 ± 0.77 a | 36.88 ± 0.85 b | 45.44 ± 1.09 a | **0.001** ** | 0.260 | **0.004** ** |
| Canopy openness (%) Aug. | 34.13 ± 1.11 a | 33.86 ± 0.90 a | 28.87 ± 0.93 b | 34.54 ± 1.52 ab | 0.195 | **0.058** . | **0.021** * |
| Canopy openness (%) Oct. | 43.59 ± 1.27 b | 43.59 ± 0.75 b | 41.67 ± 2.75 b | 52.21 ± 1.29 a | **0.032** * | **0.058** . | **0.001** ** |
| Soil Hardness | 6.04 ± 0.42 a | 7.38 ± 0.65 a | 5.88 ± 0.97 a | 7.85 ± 1.45 a | **0.055** . | 0.878 | 0.716 |
| Total litter weight (g/m²) | 89.58 ± 11.65 a | 72.21 ± 9.22 ab | 67.75 ± 4.59 ab | 41.25 ± 11.74 b | **0.072** . | **0.023** * | 0.663 |
| Humus weight (g/m²) | 123.48 ± 20.54 a | 66.60 ± 12.55 ab | 133.88 ± 33.40 a | 27.75 ± 11.58 b | **0.001** ** | 0.576 | 0.289 |
| Pieris leaf (g/m²) | 28.19 ± 3.45 a | 0.00 ± 0.00 b | 25.21 ± 2.01 a | 0.00 ± 0.00 b | > **0.001** *** | 0.694 | 0.516 |
| Bulk density (g/cc) | 0.26 ± 0.03 ab | 0.26 ± 0.02 ab | 0.18 ± 0.02 b | 0.33 ± 0.03 a | **0.077** . | 0.914 | **0.003** ** |
| Water content (g/100 cc) | 52.60 ± 2.95 a | 50.94 ± 3.00 a | 50.83 ± 2.45 a | 61.98 ± 3.48 a | 0.510 | 0.242 | **0.082** . |
| Rock (g/100 cc) | 8.43 ± 4.52 | 3.24 ± 1.86 | 0.06 ± 0.05 | 9.11 ± 7.29 | 0.835 | 0.718 | 0.111 |
| Root (g/100 cc) | 1.59 ± 0.31 | 1.16 ± 0.18 | 1.18 ± 0.23 | 0.94 ± 0.14 | 0.163 | 0.223 | 0.782 |
| pH ($H_2O$) | 4.34 ± 0.12 a | 4.28 ± 0.12 a | 4.21 ± 0.17 a | 4.68 ± 0.16 a | 0.493 | 0.381 | **0.079** . |
| EC | 0.06 ± 0.01 a | 0.06 ± 0.00 a | 0.07 ± 0.01 a | 0.06 ± 0.01 a | 0.693 | 0.382 | **0.091** . |
| Soil C (%) | 19.28 ± 1.88 a | 20.83 ± 1.85 a | 23.30 ± 2.08 a | 15.47 ± 1.52 a | 0.492 | 0.616 | **0.019** * |
| Soil N (%) | 1.04 ± 0.08 a | 1.11 ± 0.07 a | 1.26 ± 0.09 a | 0.94 ± 0.10 a | 0.577 | 0.815 | **0.030** * |
| C/N ratio | 18.09 ± 0.74 | 18.42 ± 0.83 | 18.40 ± 0.59 | 16.62 ± 0.38 | 0.637 | 0.169 | 0.112 |
| SOM content | 0.57 ± 0.03 ab | 0.59 ± 0.03 ab | 0.66 ± 0.04 a | 0.51 ± 0.02 b | 0.361 | 0.889 | **0.014** * |
| NO₃ (mg/100 g) | 0.31 ± 0.05 a | 0.31 ± 0.03 a | 0.34 ± 0.05 a | 0.45 ± 0.10 a | 0.452 | **0.078** . | 0.264 |
| NH₄ (mg/100 g) | 2.98 ± 0.20 a | 3.43 ± 0.24 a | 2.99 ± 0.43 a | 2.37 ± 0.15 a | 0.648 | **0.098** . | **0.077** . |
| K (mg/100 g) | 13.35 ± 1.74 | 15.43 ± 1.58 | 15.97 ± 5.46 | 10.84 ± 2.43 | 0.967 | 0.782 | 0.218 |
| Mg (mg/100 g) | 12.13 ± 1.12 | 12.39 ± 0.84 | 12.70 ± 1.02 | 9.33 ± 1.10 | 0.355 | 0.565 | 0.103 |

p-values:. $< 0.1$

* $< 0.05$

** $< 0.01$

and *** $< 0.001$.

Different lowercase letters indicate significance at p <0.10 determined using Tukey's test.

as litter, and particles ≤4 mm were categorized as humus. The litter and humus were oven dried at 72°C for 72 h and then weighed (± 0.1 g). After weighing the litter, the *Pieris* leaves were divided and weighed separately. The total litter, humus, and *Pieris* leaf weights were then converted into g m⁻². To quantify the soil bulk density (g cc⁻¹) and the soil water content (g 100 cc⁻¹) for each plot, soil layers were collected using a 100-cc syringes. The syringes were then oven dried at 108°C for 72 h and weighed (± 0.01 mg). The soil bulk density and soil water content were calculated using the before-and-after oven-dried weights. The rock (g 100 cc⁻¹) and plant root (g 100 cc⁻¹) contents of the syringes were divided and weighed (± 0.01 mg). To analyze the soil chemical properties, we collected topsoil under the $A_0$ horizon in plastic bags. The soil was then sieved using 2-mm mesh and air dried, To determine the soil pH ($H_2O$), the soil was extracted with water at a ratio of soil to deionized water of 1:2.5 (w/w) and the pH was measured using a soil pH tester (HI-981030; Hanna Instruments, MA, US). To determine the soil conductivity, the soil was extracted with water at a ratio of soil to deionized water of 1:5 (w/w) and the electrical conductivity (EC) was measured using a soil conductivity tester (HI-98331; Hanna Instruments). The percentage of C and N and the ratio of C to N (C/N) in the soils were analyzed using an N, C, and hydrogen analyzer with an oxygen circulating combustion system (Sumigraph NC-220F; Sumika Chemical Analysis Service, Japan) according to the manufacturer's protocol. Total soil organic matter (SOM content; g m⁻²) was calculated by soil combustion at 550°C for 12 h using a maffle furnace (FUW210PB; Advantec, Japan). Six soil nutrient contents—nitrate nitrogen ($NO_3^-$), ammonium nitrogen ($NH_4^+$),

plant-available phosphorus (P), exchangeable potassium ($K^+$), exchangeable calcium ($Ca^{2+}$), and exchangeable magnesium ($Mg^{2+}$)—were measured using a simplified soil nutrient analyzer (EW-THA1J; Air Water Biodesign Inc., Japan) according to the manufacturer's protocol, described as follows: The 1.0-g air dried soils were digested in the analyzer solution (EW-T201J; Air Water Biodesign Inc.) for 30 s and then filtered through a glass fiber filter (GA-55; Advantec). The filtered solutions were then placed in the analyzer using exclusive cartridges (EW-T102J; Air Water Biodesign Inc.) and the contents were estimated using absorbance detected by LED photo detectors. The machine analyzed the seven nutrient contents within the following ranges: $NO_3^-$: 1–50 mg 100 g dry soil$^{-1}$, $NH_4^+$: 1–50 mg 100 g dry soil$^{-1}$, P: 1–350 mg 100 g dry soil$^{-1}$, $K^+$:1–200 mg 100 g dry soil$^{-1}$, $Ca^{2+}$:1–1,000 mg 100 g dry soil$^{-1}$, and $Mg^{2+}$: 1–120 mg 100 g dry soil$^{-1}$. However, the P and $Ca^{2+}$ contents of the samples were less than the lower limits of the analyzer, therefore, we did not use the P and $Ca^{2+}$ results.

## Assessment of soil microbial communities and their functions

Soil for the microbial community analysis was collected at the same time as the soil for the chemical analysis. The soils were stored in a freezer at −20˚C until analysis. DNA was extracted using a kit (NucleoSpin Soil; Macherey–Nagel, Germany) following the manufacturer's protocol. Extracted DNA was amplified in the prokaryotic 16S rRNA V3–V4 regions using a primer set for 16S 341f and 805r, and the fungal ITS1 regions using a primer set (ITS1-F_KYO1 and ITS2_KYO2) [22] with an enzyme (KOD-FX Neo; Toyobo, Japan). The constructed libraries were sequenced on the illumina MiSeq System using paired-end 300 bp read lengths and the MiSeq Reagent Kit (ver. 3) (Illumina, CA, US) at the Bioengineering Lab. Co., Ltd. (Kanagawa, Japan). The sequence quality was checked using the FASTX-Toolkit (ver. 0.0.14), and the sequences under Q20 and shorter than 130 bases were discarded using sickle (ver. 1.33). Each paired read was united using FLASH (ver. 1.2.11). Chimera and noise were removed using the Qiime2 (ver. 2022.2) dada2 plugin, and the amplicon sequence variants (ASVs) were exported with the read count. Phylogenetic estimations of prokaryotes and fungi were conducted using a feature-classifier plugin with Greengene (ver. 13_8) and UNITE (ver. 8.3) references with a 97% threshold. Fungal functionality was estimated using FUNGuild [23].

## Identification of *Pieris* root-associated fungi

To identify root-associated fungi in *Pieris* roots, we analyzed DNA extracted from the freshly cut roots of six *Pieris* saplings (three samples were collected from each of the M01 and H03 sites). The roots were washed with sterilized distilled water three times before being sampled. We extracted the DNA using a kit (NucleoSpin Plant II; Macherey–Nagel) following the manufacturer's protocol. Fungal ITS1 amplification, library construction, MiSeq sequence analyses, raw data curation, and taxonomical and functional identifications of the read data were conducted using the same methods described in the previous section. Detailed sample information of the height, diameter at ground level ($D_0$), and raw data count of the sequences are shown in S1 Table.

## Statistical analysis

To assess whether the environmental conditions differed due to the presence of *Pieris* shrubs and the types of canopy trees, we constructed linear mixed models (LMMs) for each environmental variable. The response variables were the measured environmental variables. The fixed effects were the understory layers (*Pieris* shrubs or others), the types of canopy trees in each plot (coniferous or deciduous broad-leaved trees), and the interaction term of for these two variables. The random effect was the region (Maruju or Hirono). The significance of the fixed

effects was tested using Type II analysis of variance (ANOVA). For a *post-hoc* analysis, significant variables (p < 0.05) and marginally significant variables (p < 0.10) were tested using Tukey's multiple comparison test at the p < 0.10 threshold.

To set the read count evenly among the soil microbial community data, we rarefied the read counts for each sample of prokaryotes and fungi. The lowest number of read counts for prokaryotes was 28,592, and 36,301 for fungi. We resampled the read based on the least number for all samples and set the data matrix for the following analyses.

To compare the soil microbial alpha diversity among the sites, we calculated four alpha diversity indices (i.e., the number of ASVs, Shannon, Simpson, and inverse Simpson). The scores were analyzed using the same LMM structures described above.

To assess whether the compositions of the microbial communities differed among the sites, we calculated the Bray–Curtis dissimilarity index for all combinations of samples. The effects of *Pieris* shrubs in the understory, the types of canopy trees, and the interaction term for the similarity indices were tested using permutational multivariate ANOVA (PERMANOVA) with 10,000 permutations. We plotted the samples based on the Bray–Curtis index using nonmetric multidimensional scaling (NMDS). To analyze the relationships between the microbial community and the eight significant environmental variables detected by Tukey's test (the details are presented in the Results section), the environmental variables were tested using the envfit function, and variables with p-values <0.10 were drawn on the NMDS plot.

To assess whether microbial community taxa and fungal functionality were affected by significant environmental variables detected by the envfit functions, the relative read counts of each microbial taxon at the phylum level for both prokaryotes and fungi and fungal functionality estimated by FUNGuilds were analyzed using LMM. For the prokaryote taxa, we set the SOM content as a fixed effect because the variables represented prokaryotic community changes among the samples (the details are presented in the Results section). For the fungal taxa and two fungal functionality indices (i.e., trophic mode and guild), we set the *Pieris* leaf and litter weights as fixed effects (the details are presented in the Results section). In all LMM structures, the regions were set as random effects. All explanatory variables were normalized and tested using Type II ANOVA.

The statistical analyses were performed using R (ver. 4.1.2) [24] with the following packages: car [25], emmeans [26], lme4 [27], and vegan [28].

## Results

### Environmental properties

Canopy openness ranged from 28.9% (DB-Pieris in August) to 52.2% (DB-Other in October) throughout the research period (Table 2, S1 Appendix). The canopy trees' leaf buds were opening in June, and defoliation started in October; therefore, the effects of the understory on canopy openness were significant (Table 2; LMM: understory, p < 0.05). In both months, the DB-Pieris plot had the lowest canopy openness percentages, and the DB-Other plot had the highest percentages. In August, the general trends were the same as for June and October; however, significant *Pieris*-dominant effects were not detected. Throughout the year, the interaction terms were significant (p < 0.05); however, the *Pieris*-dominant effects were likelier to be detected in the deciduous broad-leaved tree plots rather than in the coniferous tree plots.

Total litter weight was higher in Conifer plots (LMM: canopy, p = 0.023) but did not differ by understory (LMM: understory, p = 0.072). Humus and *Pieris* leaf weights were significantly increased according to the *Pieris* dominance (LMM: understory, p = 0.001, p < 0.001, respectively). The humus weight was about five times greater in the DB-Pieris plot compared to the DB-Other plots, while the humus weights of the C-Pieris and C-Other plots were not

significantly different (Table 2). The DB-Pieris plots had significantly lower scores for soil bulk density and higher scores for SOM content compared with those of the DB-Other plots; however, the scores of the C-Pieris and C-Other plots were not significantly different (Table 2) (LMM: the interaction term, p < 0.05). For the other eight environmental variables (i.e., soil hardness, water content, pH, EC, soil C and N, $NO_3^-$, and $NH_4^+$), the understory, canopy, and interaction terms were significantly or marginally significant; however, the scores for the four plot types based on Tukey's test were not significantly different. The five variables (i.e., rock, root mass, C/N, K, and Mg) were not significantly different for any terms. For the microbial community analyses, the eight environmental variables (i.e., canopy openness over three seasons, total litter weight, humus weight, *Pieris* leaf, soil bulk density, and SOM content) that showed significant differences according Tukey's tests were used.

## Soil microbial communities and functionality

The amplicon sequencing of the prokaryotic 16S rRNA V3–V4 regions confirmed 34 phyla in two domains (archaea and bacteria) (Fig 1A, S2 Table). About 90% of the total read counts comprised by the top five phyla: Proteobacteria (average 39.5%; range 37.8%−40.6%), Acidobacteria (24.2%; 23.0%−24.8%), Actinobacteria (12.7%; 11.5%−13.7%), Planctomycetes (7.9%; 7.5%−8.3%), and Chloroflexi (4.3%; 3.1%−5.9%) (Fig 1A, S2 Table). The alpha diversity indices of the understory and canopy terms were significant (Table 3, LMM: p < 0.05), except for the number of ASVs (LMM: understory, p = 0.471). DB-Pieris had the highest average scores for all indices; however, the scores were not significantly different in the plots of deciduous broad-leaved trees, based on Tukey's test. In the fungal community, 14 phyla were confirmed by the sequencing (Fig 1B). The top five phyla, Basidiomycota (average 48.7%; range 44.9%−51.5%),

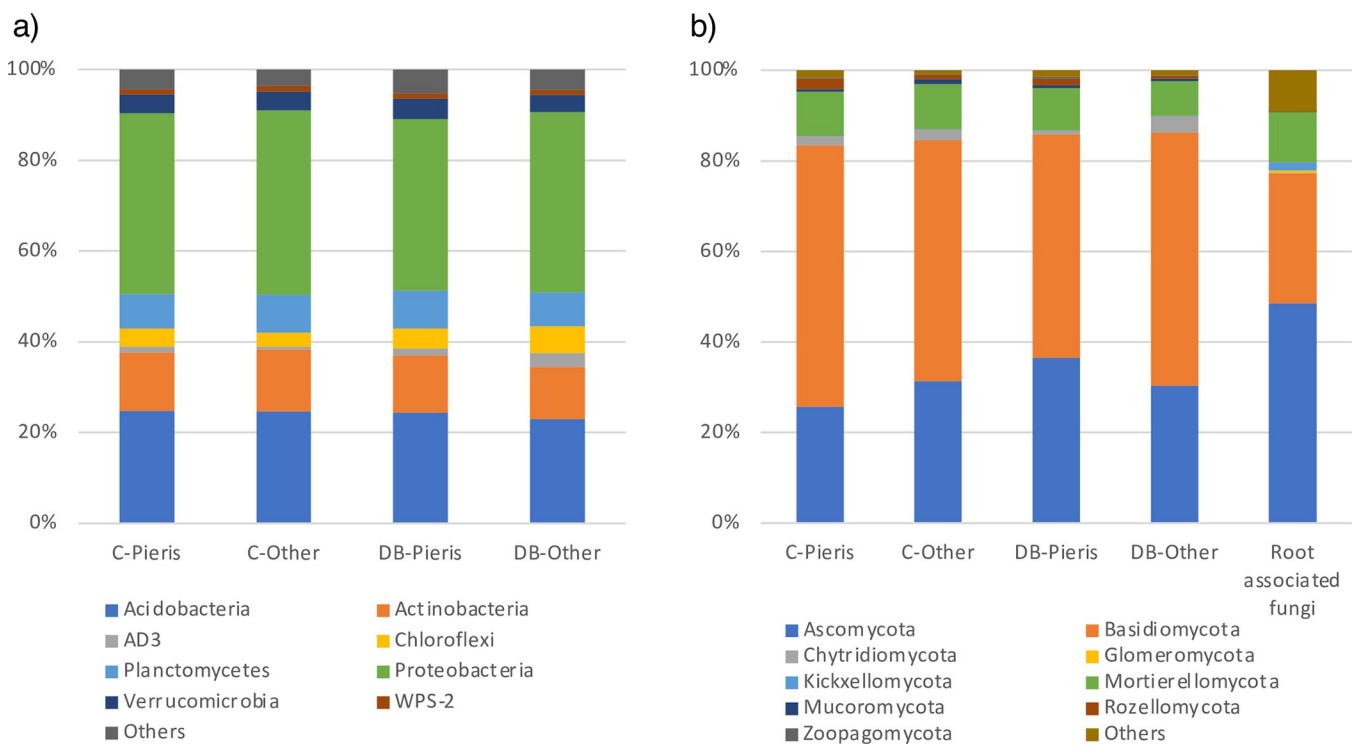

**Fig 1.** Proportions of the relative abundance of each microbial taxon: a) prokaryotes and b) fungi. In b), root-associated fungi in *Pieris* root samples are shown on the right side.

**Table 3. Prokaryotic and fungal alpha diversities for the four study site types with the results of the linear-mixed-model analyses.**

| Canopy & understory trees | | Conifers | | | | | | Deciduous broad-leaved trees | | | | | | p-values of fixed effects | | |
|---|---|---|---|---|---|---|---|---|---|---|---|---|---|---|---|---|
| | Diversity indices | Pieris | | | | Other | | Pieris | | | | Other | | Understory | Canopy | Interaction |
| Prokaryotes | Number of ASVs | 971.250 | ± | 13.142 | bc | 961.308 | ± | 21.905 | c | 1105.667 | ± | 38.777 | a | 1073.200 | ± | 25.476 | ab | 0.471 | | > 0.001 *** | 0.657 |
| | Shannon | 6.313 | ± | 0.017 | bc | 6.277 | ± | 0.020 | c | 6.455 | ± | 0.035 | a | 6.381 | ± | 0.036 | ab | 0.047 * | > 0.001 *** | 0.461 |
| | Simpson | 0.997 | ± | 0.000 | b | 0.996 | ± | 0.000 | b | 0.997 | ± | 0.000 | a | 0.997 | ± | 0.000 | ab | 0.026 * | 0.005 ** | 0.302 |
| | inverse simpson | 303.774 | ± | 9.694 | b | 287.135 | ± | 9.741 | b | 357.990 | ± | 11.662 | a | 309.843 | ± | 19.412 | ab | 0.023 * | 0.002 ** | 0.202 |
| Fungi | Number of ASVs | 208.667 | ± | 15.315 | a | 226.385 | ± | 15.196 | a | 235.500 | ± | 20.240 | a | 185.600 | ± | 17.096 | a | 0.856 | 0.652 | 0.051 . |
| | Shannon | 3.800 | ± | 0.112 | a | 4.069 | ± | 0.095 | a | 3.842 | ± | 0.134 | a | 3.926 | ± | 0.206 | a | 0.056 . | 0.438 | 0.338 |
| | Simpson | 0.935 | ± | 0.009 | | 0.951 | ± | 0.009 | | 0.926 | ± | 0.025 | | 0.947 | ± | 0.014 | | 0.126 | 0.402 | 0.949 |
| | inverse simpson | 20.709 | ± | 4.197 | | 26.854 | ± | 3.533 | | 19.330 | ± | 4.135 | | 24.935 | ± | 5.519 | | 0.141 | 0.510 | 0.847 |

p-values:. < 0.1

\* < 0.05

\*\* < 0.01, and

\*\*\* < 0.001.

Different lowercase letters indicate significance at p <0.10 determined using Tukey's test.

Ascomycota (27.9%; 22.9%−33.2%), and Mortierellomycota (8.2%; 7.0%−8.7%), constituted about 85% of the total read counts (Fig 1B, S3 Table). In general, the alpha diversity indices of the plots were not significantly different (Table 3). Two alpha diversity indices (i.e., numbers of ASVs and Shannon) were marginally significant according to the interaction term (LMM: p = 0.051) and the understory term (LMM: p = 0.056), respectively; however, the scores were not significantly different according to the understory and canopy tree types based on Tukey's test. The other two indices were not significant for any of the fixed effect terms (LMM: p > 0.10).

According to the NMDS plot based on the Bray–Curtis indices and the PERMANOVA test, prokaryotic communities differed by canopy trees (PERMANOVA: canopy, $F_{1,32}$ = 2.160, p = 0.006), while the understory and interaction terms were not significant (PERMANOVA: understory, $F_{1,32}$ = 1.109, p = 0.268; interaction, $F_{1,32}$ = 1.228, p = 0.169). The deciduous broad-leaved tree samples plotted on the right side and those of the coniferous trees plotted on the left side (Fig 2A). However, the *Pieris*-dominant site samples were plotted with other vegetation samples. In the envfit analysis, three environmental variables were significant: SOM content, litter weight, and soil bulk density (p < 0.01, Table 4) with the same or opposite arrow directions (Fig 2A). Considering the results of the LMM for the environmental variables (Table 2) and the correlation coefficients among the environmental variables (S4 Table), these three variables roughly divided the samples into two groups: DB-Other (i.e., low SOM content and litter weight and high soil bulk density) and C-Pieris, C-Other, and DB-Pieris (i.e., high SOM content and litter weight and low soil bulk density). Since the $R^2$ values of the SOM content were highest among the three environmental variables (Table 4), we constructed an LMM to assess which prokaryotic taxon changed with SOM content. SOM content positively affected the relative abundances of Actinobacteria, Verrucomicrobia, Chlamydiae, and the other nine phyla, and it negatively affected the relative abundance of Chloroflexi (S2 Table, S2 Fig).

In the fungal communities, (Fig 2B), the understory vegetation and canopy tree type terms were significant (PERMANOVA: understory, $F_{1,32}$ = 1.596, p < 0.001; canopy, $F_{1,32}$ = 1.367, p = 0.008). Since the interaction term was not significant (PERMANOVA: $F_{1,32}$ = 1.036, p = 0.320), fungal community changes due to *Pieris* dominance were the same in coniferous and deciduous broad-leaved tree stands. In the NMDS plots, *Pieris*-dominant plots plotted on the top left side, and the other understory vegetation plotted on the opposite side (Fig 2B). Of the analyzed environmental variables, six environmental variables (i.e., *Pieris* leaf, humus

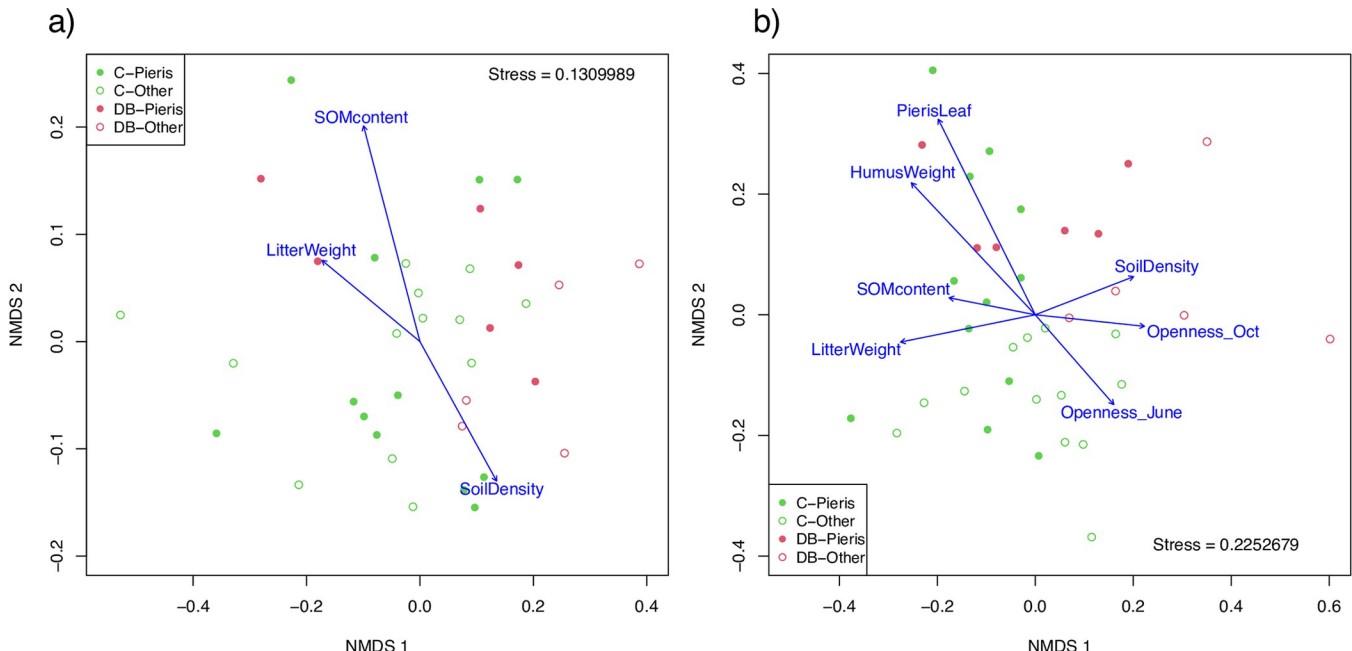

**Fig 2.** Nonmetrical dimension scaling plot of two microbiomes based on Bray–Curtis indices and significant environmental variables detected via the envfit function (see Table 3, p < 0.10) shown as arrows: a) prokaryotes and b) fungi. The green dots indicate the samples from conifer trees, and the red dots indicate samples from deciduous broad-leaved trees. The solid dots indicate samples from understories dominated by *Pieris*, and the open dots indicate samples from under other understory vegetation.

weight, litter weight, canopy openness in October and June, and soil bulk density, in the order of the R$^2$ values) significantly affected the fungal communities, and SOM content marginally affected (p = 0.099) (Table 4). Considering the directions of the environmental variable arrows shown in the NMDS plot and the correlation coefficients among the environmental variables (S4 Table), the effects of the environmental variables on the fungal communities were divided into two directions: *Pieris*-dominant effects (i.e., high *Pieris* leaf and humus weights and low

**Table 4. Effects of the investigated environmental variables on prokaryotic and fungal communities analyzed using the envfit function for the nonmetric multidimensional scaling (NMDS) analysis.**

| | Prokaryotes | | | | | Fungi | | | | |
|---|---|---|---|---|---|---|---|---|---|---|
| | NMDS1 | NMDS2 | R$^2$ | p-values | | NMDS1 | NMDS2 | R$^2$ | p-values | |
| Openness June | 0.301 | 0.954 | 0.022 | 0.703 | | 0.733 | -0.680 | 0.197 | **0.031** | * |
| Openness Aug. | 0.758 | 0.652 | 0.017 | 0.760 | | 0.891 | -0.454 | 0.003 | 0.947 | |
| Openness Oct. | 0.524 | 0.851 | 0.048 | 0.450 | | 0.996 | -0.085 | 0.207 | **0.023** | * |
| Litter weight | -0.916 | 0.402 | 0.300 | **0.003** | ** | -0.987 | -0.163 | 0.320 | **0.002** | ** |
| Humus weight | -0.720 | -0.694 | 0.059 | 0.363 | | -0.757 | 0.654 | 0.461 | **>0.001** | *** |
| Pieris leaf | 0.646 | 0.764 | 0.001 | 0.985 | | -0.521 | 0.853 | 0.594 | **>0.001** | *** |
| Bulk density | 0.721 | -0.693 | 0.297 | **0.004** | ** | 0.954 | 0.300 | 0.183 | **0.034** | * |
| SOM content | -0.443 | 0.896 | 0.425 | **>0.001** | *** | -0.987 | 0.158 | 0.131 | **0.099** | . |

p-values:. < 0.1

* < 0.05

** < 0.01, and

*** < 0.001.

canopy openness) and specific effects in DB-Other (i.e., low litter weight and SOM content and high soil bulk density). According to the $R^2$ values from the envfit results, the former effect was represented by *Pieris* leaf ($R^2 = 0.594$), and the latter was represented by litter weight ($R^2 = 0.320$). We then assessed the effects of *Pieris* leaf and litter weight on the relative abundance of fungal phylum and the functionalities using an LMM. Two fungal phyla (i.e., Calcarisporiello-mycota and Rozellomycota) were positively affected by *Pieris* leaf, and Mortierellomycota was positively affected by litter weight (S3 Table). For fungal functionality, the trophic mode was generally affected by *Pieris* leaf (Fig 3, S5 Table). The relative abundance of symbiotroph

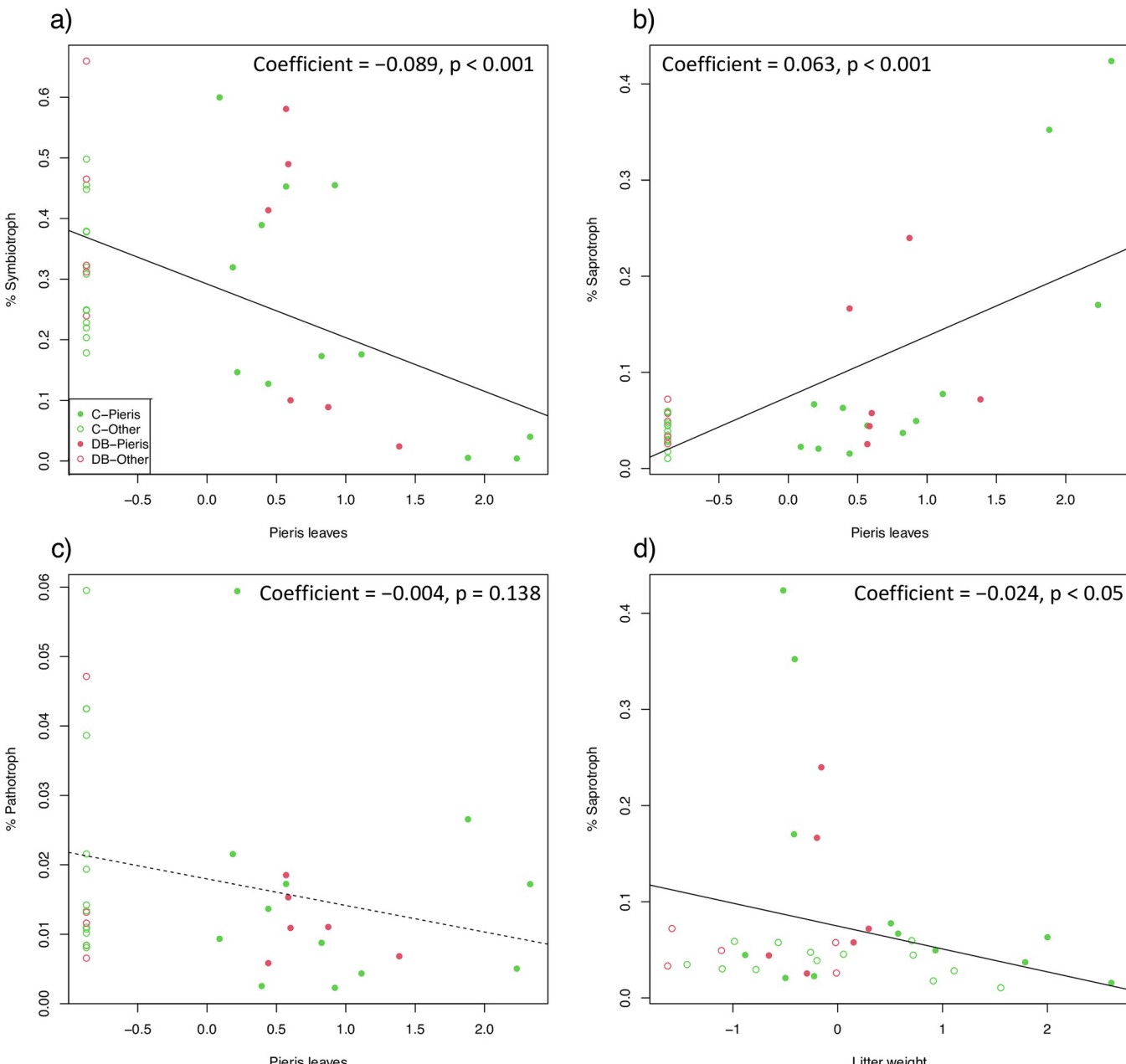

**Fig 3.** Relationships between significant environmental variables and the relative abundance of fungal trophic modes: a) weight of *Pieris* leaf and the relative abundance of symbiotrophs, b) saprotrophs, and c) pathotrophs, and d) total litter weight and saprotrophs. The explanatory variables were standardized. The solid line indicates the significant effect of the explanatory variable, and the dotted line indicates that no significant effects were detected (see also S6 Table).

decreased (p < 0.001, Fig 3A) and that of saprotroph increased with *Pieris* leaf (p < 0.001, Fig 3B). Pathotrophs were not affected by either *Pieris* leaf or litter weight (*Pieris* leaf, p = 0.13, Fig 3B; litter weight, p = 0.98). Litter weight negatively affected the relative abundance of saprotroph (p < 0.05, Fig 3D). At the guild level, the relative abundance of ECM fungi decreased with *Pieris* leaf (p < 0.001, S6 Table, S3 Fig). However, other symbiotic fungi, such as endophyte, ErM, and AM, were not affected by *Pieris* leaf (p > 0.10, S6 Table, S3 Fig).

### *Pieris* root-associated fungi

Root-associated fungi of *P. japonica* have been confirmed by microscope observations [15]; however, molecular identification and host ranges have not been determined. Nine fungal phyla were confirmed in the six *Pieris* root samples by the amplicon sequencing (Fig 1B). The top three phyla, Ascomycota (average 48.5%; range 20.0%−78.3%), Basidiomycota (28.8%; 6.4%−63.0%), and Mortierellomycota (11.0%; 0.6%−45.3%), comprised about 90% of the total read counts (Fig 1B, S3 Table). At the order level, 48 orders were confirmed, and known ErM taxa [12], such as Helotiales (average 22.9%), Sebacinales (average 11.0%), Agaricales (average 5.84%), and Hypocreales (average 1.56%) were confirmed in all samples (S7 Table). Chaetothyriales (average 3.42%) and Trechisporales (average 0.11%), including ErM and the dark septate endophyte, an important endophyte found in Ericaceae plant roots, were found in some samples. ECM taxa—Cantharellales (average 6.50%), Thelephorales (average 2.75%), Russulales (average 1.66%), Atheliales (average 0.29%), Auriculariales (average 0.25%), and Boletales (average 0.12%)—and an AM taxon, Glomeromycota—Glomerales (average 0.57%) and Diversisporales (average 0.002%)—were also confirmed from some samples.

The functionalities of root-associated fungi were estimated. In the trophic mode, saprotroph (average 14.1%), symbiotroph (average 13.0%), and saprotroph–symbiotroph (average 12.5%) were frequently confirmed (S5 Table). Pathotroph was only confirmed by 2.21%, on average. At the guild level, undefined saprotrophs (average 12.0%) and multifunctional saprotrophs, such as endophyte-litter saprotroph-soil saprotroph-undefined saprotroph (average 11.0%), were confirmed as frequent guilds (S6 Table).

## Discussion

### Differences between understory environments according to *Pieris* dominance for different canopy trees

The understory environments differed according to the types of understory and canopy trees. Under deciduous broad-leaved trees, *Pieris* dominance led to high *Pieris* leaf litter and humus weights and low soil bulk density and canopy openness. These changes have also been observed in Ericaceae monodominant areas [e.g., 2, 29–31]. DB-Pieris had the lowest canopy openness percentages compared with other plots in the same season, whereas those of DB-Other seasonally fluctuated probably responding to the canopy trees' leaf phenology, that is, leaf buds opening (June), leaves matured (August), and defoliation started (October). *Pieris*-dominances in understory might make the light environments less variable throughout the year. Due to the formation of the understory canopy, the light quantity and quality are often affected. Under the canopy of *Rhododendron maximum* (Ericaceae), photosynthetically active radiation decreases compared with that in nonrhododendron areas [32]. Messier et al. [29] reported the decreased red:far-red ratio in the understory canopy of *Gaultheria shallon* (Ericaceae), which can affect the seed germination rate. Along with the formation of the understory canopy, the light environment on the forest floor would become less variable [32, 33], and thus, *Pieris* dominance in the forest understory may change both the quantity and quality of

the light environment. Thick layers of humus formed by *Pieris* dominance also impact the forest ecosystems and often physically prevent the establishment of seedlings, which inhibits forest regeneration [30, 34]. Ericaceae humus and roots contain many secondary compounds, such as phenolic acid and tannin, and extracellular enzymes, and the decomposition rates of the humus are slow [7, 35, 36]. These changes in chemical contents and the decomposition rate affect seed germination and growth and N cycles derived from the secondary compounds of Ericaceae plants [6–8, 37]. Moreover, chemical inhibition is often dependent on combinations of Ericaceae species and non-Ericaceae plants [38, 39]. A few individual seedlings and saplings of other-trees were observed in the *Pieris*-dominant forest [3, Tokumoto, Y. unpublished data]. Although an allelopathic test of the *Pieris* leaf on lettuce seeds showed no significant effect on seed germination [40], the effects of *Pieris* root, humus, and leaf litter have not been tested. Therefore, some chemical inhibition of *Pieris* species may exist; however, the inhibition process is not well understood. Regarding the nutrient cycles, our soil chemical analyses on soil N and C and other nutrient ions showed that nonsignificant changes were inflicted by *Pieris* dominance (Table 2), and the impact of *Pieris* dominance on nutrient cycles may be less than that previously reported for other Ericaceae plants [37]. Because the other variables, such as soil nitrogen complexes and enzyme activities, have not been measured in this study, further studies on the physical and chemical factors that inhibit plant regeneration under *Pieris* populations and the effects on the nutrient cycles in forest ecosystems are required to better understand the effects of *Pieris* dominance.

In this study, most of the effects of *Pieris* dominance were observed in deciduous broad-leaved tree stands rather than in evergreen coniferous stands, except for the amount of *Pieris* leaf. In particular, DB-Other had the highest soil bulk density scores and the lowest litter weights, which is similar to the results for soil-eroded regions in the same study regions (soil bulk density: 0.29–0.40 g/cm$^3$) [41]. After understory layers have been grazed by deer, litter and soils are frequently removed, and high soil bulk density and low chemical contents, such as SOM, C, and N, are often observed in severely eroded regions [41–43]. At our research sites, surface soils were almost exposed under deciduous broad-leaved trees; therefore, the impact of direct rainfall on the soil surfaces of deciduous broad-leaved tree stands will be greater than that in evergreen coniferous stands. This implies that *Pieris* cover has prevented soil erosion in DB-Pieris; and thus, *Pieris* cover in the understory might play a role in preventing litter and soil runoff in deciduous broad-leaved tree stands and maintaining the soil's physical properties and chemical contents after deer grazing. However, no significant effect of *Pieris* on soil properties was observed in the coniferous stands, which suggests that the effect of *Pieris* cover differs among forest types. Our study region has experienced severe deer overgrazing, and the original understory vegetation, especially dwarf bamboo, has disappeared from many areas. After the disappearance of understory vegetation, the invasion of thickets of *Pieris* may remediate forest soil environments.

### Differences between soil microbial communities according to *Pieris* dominance for different canopy trees

The types of understory and the canopy trees affected the microbial community compositions, diversity indices, and functionalities. *Pieris* affected the fungal community compositions but did not affect the prokaryotic community compositions. The effects of ErM plants and fungi on vegetation cover, SOM, and C and N cycles have been assessed [7]; however, research on soil microbial communities is limited. In this study, the effects of *Pieris* dominance determined by the environmental variables on the microbial communities and *Pieris* root-associated fungi are discussed below.

For the soil fungal community, the community changes were associated with the environmental changes in two directions: *Pieris*-dominant effects (i.e., high *Pieris* leaf and humus weights and low canopy openness) and specific effects in the DB-Other plots (i.e., low litter weight and SOM content and high soil bulk density) (Fig 2B, Table 3). For the *Pieris*-dominant effects, the relative abundance of symbiotroph, especially ECM, decreased, and those of saprotroph increased with the increase of *Pieris* leaf, indicating that *Pieris* dominance led to symbiotroph-poor and saprotroph-enriched fungal communities in the soils (Fig 3A, 3B, S5 and S6 Tables). In *Pieris* root, we confirmed the association of not only ErM, AM, and endophyte but also ECM (Fig 3A, S3 Fig., S6 and S7 Tables), although previous microscopic observations of *Pieris* roots have only confirmed associations with ErM and AM [15]. Multiple types of mycorrhizae fungi have been found in the roots of other ericaceous plants using molecular identification (e.g., ECM, ErM, and AM in *Pyrola japonica* [44]; ErM and AM in *Vaccinum oldhamii* [45]). Our molecular identification of root-associated fungi suggests that *P. japonica* may potentially have broad mutualistic relationships with three mycorrhizae. However, in the soil fungal communities, the proportions of symbiotroph, particularly ECM fungi were negatively affected by *Pieris* dominance, while other fungi, such as ErM, AM, and endophyte, were not significantly affected (S5 Table, S3 Fig). Some studies showed that the ErM fungi commonly live in non-Ericaceae plant roots and these plants may maintain the ErM fungi populations [46]; therefore, *Pieris* dominance might not affect the relative abundance of ErM in soils. In contrast, the result of the decrease of proportions of ECM fungi was inconsistent with the previous study [47] that showed the proportions of ECM fungi increased to the amount of leaf litter inputs. The negative proportional changes of ECM fungi associated with the *Pieris* leaf could be discussed with the interactions between the Ericaceae plants or ErM. Previous studies have reported that the infection rate of ECM in plants, which have the symbiotic relationships with ECM, was decreased by Ericaceae plants in pot experiments, and that Ericaceae and ErM might suppress ECM fungal infection of plant roots [9, 10]. Although some studies have reported that ErM plant cover is positively related to the abundance of ECM plants at plot levels [48], the other study also showed that the directions and intensities of ErM effects have differed among combinations of the ErM and ECM plant species [10]. In our study regions, *Pieris* shrubs or ErM fungi infected in *Pieris* may suppress the ECM in soils, however, the underlying mechanisms on how the proportions of ECM decreased were unknown. Further studies are needed to understand these mechanisms and the interactions between ECM and *Pieris* shrubs or ErM.

*Pieris*-dominant effects were also confirmed by changes in the proportions of saprotrophic fungi; their increased proportions were associated with *Pieris* dominance (Fig 3B, S4 Fig). *Pieris* dominance in the understory leads to more undecomposed organic matter, such as humus layers and *Pieris* leaf, and less canopy openness (Table 1). This positive proportional change of saprotrophic fungi might relate to such environmental changes and the result was consistent with that of the previous study, which tested the soil microbial community changes according to the leaf litter input [47]. According to the other previous study, ErM fungi also show high saprotrophic capability and have been predicted to compete with free-living saprotrophs [7]. In our results, the proportional changes in ErM were not confirmed, and the number of other saprotrophic fungi may have increased in response to increased organic matter in the *Pieris*-dominant areas without the distinctive competition with ErM. In contrast, litter weight negatively affected the proportions of saprotroph (Fig 3D), which indicated that DB-Other, where the litter weight was the lowest among the sites, had relatively higher proportions of saprotrophs. Compared with the evergreen coniferous forest, the topsoil of the deciduous broad-leaved tree forests has high proportions of saprotrophs [49]. In DB-Other, deciduous broad-leaved trees, such as *Quercus* and *Carpinus* spp. are distributed, and the

higher proportions of saprotroph in the plots may be due to the forest type. This suggests that the formation of *Pieris* cover may have changed the original diverse fungal communities, including saprotrophs, in different forest types into the similar soil fungal communities. Changes in soil fungal compositions sometimes affect the percentages of ECM fungal root colonization and decrease other plants' growth [9, 10]. Because few seedlings were observed under the *Pieris* canopy, suppression effects of the changed fungal community due to *Pieris* cover could be expected. Together with the effects of environmental changes caused by *Pieris* dominance, the effects of soil microbial community changes induced by *Pieris* dominance on the plant performances should also be assessed. Moreover, previous studies have reported that ErM consistently inhabits after Ericaceae plants are cut back [6, 50]. This indicates that *Pieris*-dominant effects can remain even after *Pieris* shrubs are cut back. Further studies on long-term plant performance, including germination and growth, should be conducted to understand plant–microbe interactions under *Pieris* understory canopies.

Similar to the fungal changes associated with litter weight, the prokaryotic community was also affected by SOM content, which was significantly positively correlated with litter weight and negatively correlated with soil bulk density (Fig 2A, Table 3). Proportions of the relative abundances of Actinobacteria and Verrucomicrobia changed positively with SOM and *vice versa* for Chloroflexi (S2 Fig). DB-Other had the lowest SOM content, and thus, the prokaryotic flora in DB-Other would have become less organic matter-dependent communities compared to the other three (i.e., DB-Pieris, C-Pieris, and C-Other). The above-mentioned phyla often respond to organic matter and soil depth [51, 52], and Chloroflexi is an oligotrophic bacterium that inhabits poor-nutrient soils [53]. In the previous study around our study regions, the severe soil erosion after deer overgrazing caused the soil ecosystems to change into organic matter-poor environments and oligotrophic bacterium-enriched prokaryotic communities [54]. As mentioned, the DB-Other soil environments closely resembled those of soil-eroded areas in terms of the litter weight, bulk density, and SOM content. In the absence of *Pieris* shrubs, soil prokaryotic flora might resemble those of deeper and poor-nutrient soils, especially in deciduous broad-leaved tree stands. A previous study has compared the extracellular enzyme activities between sites with or without Ericaceae removal, and the presence of Ericaceae decrease C-acquisition enzymes, such as β-glucosidase and β-xylosidase and soil C and N availabilities [55]. Although the soil chemical parameters and the soil prokaryote community in our study were not directly affected by the *Pieris*-dominances (Tables 2 and 4), a further study on the microbial activities, including enzyme activities is important to investigate in details the effects of the *Pieris*-dominances on the ecosystem.

## Conclusions

This study focused on the understory environments and soil microbiomes of *Pieris*-dominant sites with two different types of upper tree stands. Due to the *Pieris* dominance in the understory, canopy openness decreased by the formed understory canopy and humus layers were thickened, especially in the deciduous broad-leaved tree stands. The soil chemical parameters, which were often changed by other Ericaceae plant dominance in the understory layer, were not different between *Pieris*-dominant and nondominant sites. However, soil fungal communities were affected by the *Pieris* dominance, and the proportions of symbiotrophs, especially ECM fungi, decreased, and those of pathotrophs increased, although prokaryote communities were not affected by the *Pieris* dominance. These changes in the environmental variables and soil microbial communities might affect forest regeneration and, indeed, few seedlings were observed under the *Pieris* populations. The results should provide fundamental knowledge of the effects of *Pieris* monodominant in the forest understories and will encourage future studies

on the forest ecosystems and mountain protections in changing environments after deer overgrazing.

## Supporting information

**S1 Fig. Location of the research regions (Maruju, M; Hirono, H).** *Pieris* dominated sites were shown with odd numbers (M01, M03, M05, H01, H03, H05, and H07). Sites with other vegetation (sites with even numbers) were located near the sites with odd numbers.
(TIFF)

**S2 Fig.** Relationships between SOM content and the relative abundance of prokaryotic phyla: a) Actinobacteria, b) Verrucomicrobia, c) Chlamydiae, and d) Chloroflexi. The explanatory variables were standardized, and the results are shown as a solid line (also see S2 Table).
(TIFF)

**S3 Fig.** Relationships between the weight of *Pieris* leaves and the relative abundance of fungal symbiotrophic guilds: a) ectomycorrhizal fungi, b) endophyte, c) ericoid mycorrhizal fungi, and d) arbuscular mycorrhizal fungi. The explanatory variables were standardized. The solid line indicates the significant effect of the explanatory variable, and the dotted line indicates no significant effects (also see S6 Table).
(TIFF)

**S4 Fig. Relationships between the weight of *Pieris* leaves and the relative abundance of undefined saprotroph.** The explanatory variables were standardized, and the results are shown as a solid line (also see S6 Table).
(TIFF)

**S1 Table. *Pieris* sapling information for assessing root-associated fungi.**
(XLSX)

**S2 Table. Results of the linear-mixed-model analyses of the relative abundance of prokaryotic phyla with SOM content.**
(XLSX)

**S3 Table. Results of the linear-mixed-model analyses of the relative abundance of fungal phyla with two environmental variables and the read count (%) of root-associated fungi in the *Pieris* root samples.**
(XLSX)

**S4 Table. Results of Spearman's correlation coefficient test for the environmental variables.** The upper and lower parts of the triangular matrix are the correlation coefficient and p-value, respectively.
(XLSX)

**S5 Table. Results of the linear-mixed-model analyses of the relative abundance of fungal trophic mode with two environmental variables and the read count (%) of root-associated fungi in the *Pieris* root samples.**
(XLSX)

**S6 Table. Results of the linear-mixed-model analyses of the relative abundance of fungal guild with two environmental variables and the read count (%) of root-associated fungi in the *Pieris* root samples.**
(XLSX)

**S7 Table. Relative read count of the order-level *Pieris* root-associated fungi of six *Pieris* root samples.**
(XLSX)

**S1 Appendix. Environmental data measured in this study.**
(XLSX)

## Acknowledgments

We are grateful to the Shiiba Research Forest of the Department of Agro–environmental Sciences, Kyushu University, for their kind support during the research.

## Author Contributions

**Conceptualization:** Yuji Tokumoto, Ayumi Katayama.

**Data curation:** Yuji Tokumoto.

**Formal analysis:** Yuji Tokumoto.

**Funding acquisition:** Yuji Tokumoto, Ayumi Katayama.

**Investigation:** Yuji Tokumoto.

**Methodology:** Yuji Tokumoto, Ayumi Katayama.

**Project administration:** Yuji Tokumoto.

**Resources:** Yuji Tokumoto, Ayumi Katayama.

**Software:** Yuji Tokumoto.

**Supervision:** Yuji Tokumoto.

**Validation:** Yuji Tokumoto.

**Visualization:** Yuji Tokumoto.

**Writing – original draft:** Yuji Tokumoto.

**Writing – review & editing:** Yuji Tokumoto, Ayumi Katayama.

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
