## [Decision Letter · Decision Letter 0]

28 Sep 2023

PONE-D-23-27922Effects of Pieris japonica (Ericaceae) dominance on cool temperate forest altered-understory environments and soil microbiomes in Southern JapanPLOS ONE

Dear Dr. Tokumoto,

Thank you for submitting your manuscript to PLOS ONE. After careful consideration, we feel that it has merit but does not fully meet PLOS ONE’s publication criteria as it currently stands. Therefore, we invite you to submit a revised version of the manuscript that addresses the points raised during the review process.

We look forward to receiving your revised manuscript.

Kind regards,

Jian Liu

Academic Editor

PLOS ONE

Journal Requirements:

Additional Editor Comments:

The study is interesting while the manuscript has some problems as suggested by the two reviewers. The authors should respond to the comments of the reviewers one by one and revise the manuscript accordingly. The revised manuscript would be sent to the reviewers for further reviewing.

Reviewers' comments:

Reviewer's Responses to Questions

**Comments to the Author**

1. Is the manuscript technically sound, and do the data support the conclusions?

Reviewer #1: Partly

Reviewer #2: Partly

2. Has the statistical analysis been performed appropriately and rigorously? 

Reviewer #1: No

Reviewer #2: Yes

3. Have the authors made all data underlying the findings in their manuscript fully available?

Reviewer #1: Yes

Reviewer #2: Yes

4. Is the manuscript presented in an intelligible fashion and written in standard English?

Reviewer #1: No

Reviewer #2: Yes

5. Review Comments to the Author

Reviewer #1: This study compared the differences between Pieris-dominant sites and Pieris-nondominant sites in the understory environment (environment, soil physicochemical properties, and microbial communities). The main advantages and disadvantages of Pieris dominance in the forest understory due to overgrazing of deer were explored. The setting of the research area and sampling points is reasonable, but there are big problems in data analysis and research conclusions. In this study, the data support of the two important conclusions(1. Pieris-dominant inhibit the establishment of forest regeneration; 2. Pieris may play an important role in preventing soil erosion) is insufficient, and they are both speculations without experimental data.

This study carried out analysis of the above-ground environment of the forest, soil physicochemical properties, root fungi, and soil bacterial and fungal communities, with a large amount of data, but few reliable conclusions have been obtained. It is suggested that the author should focus on the analysis of microorganisms with a large amount of data. Based on the analysis and function prediction of microbial communities, the author can predict the impact of Pieris-dominant on forest ecological functions from the perspective of microorganisms, and analyze the causes of microbial community changes by combining physical and chemical property data and forest environmental data.

There are so many bacteria shown in Figure 1a that some of the colors of the legend seem difficult to distinguish. You can show only the first few microorganisms with relatively high abundance, and the rest are represented by “Others”.

Line 283-285 This sentence is too wordy, please rephrase it.

Fig. 2 Why are the NMDS plots of the two microbiomes based on the Chao index? As far as I know the NMDS was usually performed based on the Bray-Curtis dissimilarity of bacterial communities.

Line313: Fig. 2b?

Reviewer #2: In this study, the difference in canopy openness, soil property and microbiome was compared between different forests with Pieris japonica after deer overgrazing. The authors found that this shrub lead to more litter and humus in deciduous broad-leaved forests. It also related to low soil bulk density and canopy openness. Moreover, it changes fungal functional group. The author gave some suggestions to forest management. However, some questions should be answered before publication.

Line 107, “Due to shika deer overgrazing”; Line 410, “Our study region has experienced severe deer overgrazing”. What is the population size or density of deer in the study site? Maybe the author can cite other study.

In Material and Methods, what are the soil types in different forests?

In Results, it is suggested to divide Environmental properties into two paragraph, e.g. canopy openness, and soil physical and chemical properties.

In Line 276, 34 phyla were confirmed in two kingdoms, however, there are only top five phyla and 14 phyla in fungal community, what are the rest phyla?

In Line 277, are the top five phyla belong to prokaryote? How many phyla in prokaryote?

In Discussion, there is no contents about canopy openness in the first paragraph, only soil properties. It is related to soil temperature, soil water and microbiome activity.

Line 431-432, Multiple types of mycorrhizae fungi have been found in the roots of other ericaceous plants using molecular identification (e.g., Matsuda et al. 2012; Baba et al. 2016). Invalid citation. What are the main types of mycorrhizae fungi in these studies?

In Conclusions and suggested future studies on Pieris-dominant forests, it is suggested to move future studies into Discussion. Conclusion need to be focused on soil change.

Line 488-489, A drawback of Pieris-dominant areas is that they inhibit the establishment of fore regeneration. Which table or figure support this conclusion? There is no direct result.

Line 498-499, A potential advantage of Pieris is that it may play an important role in preventing soil erosion, in terms of soil bulk density and subsequent prokaryotic communities. Please focus on your findings. You did not measure soil erosion.

6. PLOS authors have the option to publish the peer review history of their article (what does this mean?). If published, this will include your full peer review and any attached files.

Reviewer #1: No

Reviewer #2: No

---

## [Author Response · Author response to Decision Letter 0]

20 Nov 2023

Response to the academic editor and reviewers’ comments on Ms. No. PONE-S-23-32909 Effects of Pieris japonica (Ericaceae) dominance on cool temperate forest altered-understory environments and soil microbiomes in Southern Japan

Response to the academic editor’s comments

Dear Professor, Dr. Jian Liu, Academic Editor of PLOS ONE,

Thank you very much for your review of our manuscripts.

We would like to respond to your comments below. 

Journal Requirements:

[Res] We have amended all documents accordingly.

[Res] We deleted the funding information for this research from the manuscript file.

[Res] We have uploaded the raw sequence data to the repository and obtained the accession numbers. All the raw sequence data of the soil prokaryotic 16S rDNA, the soil eukaryotic 18S rRNA genes, and the root-associated fungal 18S rRNA genes were submitted in the Sequence Read Archive of DDBJ database under the accession number DRA016984, DRA016985, and DRA016986, respectively.

Environmental data used in the analyses were included in the Supporting information file as an Appendix1.

[Res] We wrote the statement at the end of the study site in MM section.

[Res] We added the captions for SI at the end of the file.

Additional Editor Comments:

The study is interesting while the manuscript has some problems as suggested by the two reviewers. The authors should respond to the comments of the reviewers one by one and revise the manuscript accordingly. The revised manuscript would be sent to the reviewers for further reviewing.

[Res] Thank you very much for your comments and for giving us the opportunity to improve the manuscript. We amended the manuscript files based in the reviewers’ comments. 

During the revised preparations, we found several mistakes on the former files and we corrected the mistakes as follows:

- In Table 2, the unit of soil bulk density was corrected from (g/100cc) to (g/cc).

- The sampling sites of saplings for the Pieris root-associated fungi were corrected from M02 and H02 to M01 and H03 (L194).

We would like you to confirm the documents again and would be grateful if the revised manuscript would meet the standard for the publication of PLOS ONE.

Sincerely,

Yuji Tokumoto, on behalf of the authors

 

Response to the reviewers’ comments

Dear Reviewer #1:

We appreciate the critical and constructive comments on our manuscript. We would like to respond to your comments below.

Reviewer #1: This study compared the differences between Pieris-dominant sites and Pieris-nondominant sites in the understory environment (environment, soil physicochemical properties, and microbial communities). The main advantages and disadvantages of Pieris dominance in the forest understory due to overgrazing of deer were explored. The setting of the research area and sampling points is reasonable, but there are big problems in data analysis and research conclusions. In this study, the data support of the two important conclusions(1. Pieris-dominant inhibit the establishment of forest regeneration; 2. Pieris may play an important role in preventing soil erosion) is insufficient, and they are both speculations without experimental data.

This study carried out analysis of the above-ground environment of the forest, soil physicochemical properties, root fungi, and soil bacterial and fungal communities, with a large amount of data, but few reliable conclusions have been obtained. It is suggested that the author should focus on the analysis of microorganisms with a large amount of data. Based on the analysis and function prediction of microbial communities, the author can predict the impact of Pieris-dominant on forest ecological functions from the perspective of microorganisms, and analyze the causes of microbial community changes by combining physical and chemical property data and forest environmental data.

[Res] Thank you very much for pointing out the issues of our manuscripts. In response, we checked the results and discussions again and modified the manuscript as follows. 

 For the impact of Pieris-dominant on forest ecological functions, we added the light environmental changes and its potential impacts on other plants’ performance, such as germination, in the discussion (L406-414 in the cleaned manuscript file). For other parameters, Ericaceae dominances often affect the soil N and C content and nutrient cycles according to previous studies. In this sense, we did not identify the changes in soil chemical parameters by the Pieris dominance. We just introduced the previous studies about the chemical parameters and the weak possibilities of nutrient cycle changes by the Pieris dominance for future studies (L425-430).

 For the fungal community, we reviewed the previous studies and rearranged the discussion and former conclusions. Firstly we discussed the symbiotrophic fungal changes together with the root-associated fungi and the factors which would induce the symbiotrophic fungal proportional changes (L481-494). Secondly, we mentioned the proportional changes of saprotrophic fungi and modified the previous discussions (L499-519). At the end of the discussion, we mentioned the impacts of fungal community changes on other plants, and, for further studies, we modified the sentences of the former discussions and conclusions (L519-529).

For the prokaryote community, the factors of prokaryote community changes have been discussed partially in former discussions, but, we added the recent studies including our research team’s results about the prokaryote community changes with soil erosional environmental changes (Chen et al. 2023). Actually, the soil erosion around our research site has been observed mostly because of the dissapperance of understory trees. We also modified the sentences and paragraph (L530-546).

There are so many bacteria shown in Figure 1a that some of the colors of the legend seem difficult to distinguish. You can show only the first few microorganisms with relatively high abundance, and the rest are represented by “Others”.

[Res] We formatted Figure 1a showing major 8 phyla that had more than 1% of total read counts and “Others” for the rest. Figure 1b is also an indistinguishable figure, and we showed the top 9 taxa and the rest are shown as ”Others”.

Line 283-285 This sentence is too wordy, please rephrase it.

[Res] We shortened the former sentence and rephrased the sentence as below: “DB-Pieris had the highest average scores for all indices; however, the scores were not significantly different in the plots of deciduous broad-leaved trees, based on Tukey’s test.” (L284-286)

Fig. 2 Why are the NMDS plots of the two microbiomes based on the Chao index? As far as I know the NMDS was usually performed based on the Bray-Curtis dissimilarity of bacterial communities.

[Res] We have used the Chao index because the indices considered the rare or unseen species. However, according to your comments and the previous studies re-reviewed, the index has not been used in the microbial community data. So, to respond to this comment, we re-analysed the microbiome data based on the Bray-Curtis dissimilarity. We redrew the NMDS plot and conducted envfit analysis and PERMANOVA. Some results were the same trends as the previous results, and some results were completely the same as the previous results (Fig. 2., Table 4, Section Results). 

Line313: Fig. 2b?

[Res] Fig. 2b was incorrect and it should be 1b. We amended it. Thank you very much.

We hope our responses could meet your criterion for evaluation and revision.

Sincerely,

Yuji Tokumoto

 

Dear Reviewer #2:

Thank you very much for the review and positive and constructive comments. We would like to respond to the comments below.

Reviewer #2: In this study, the difference in canopy openness, soil property and microbiome was compared between different forests with Pieris japonica after deer overgrazing. The authors found that this shrub lead to more litter and humus in deciduous broad-leaved forests. It also related to low soil bulk density and canopy openness. Moreover, it changes fungal functional group. The author gave some suggestions to forest management. However, some questions should be answered before publication.

Line 107, “Due to shika deer overgrazing”; Line 410, “Our study region has experienced severe deer overgrazing”. What is the population size or density of deer in the study site? Maybe the author can cite other study.

[Res] There was a previous study in this study site (but in Japanese), and we added the population density in the text (L94-95 in the cleaned manuscript file). From 1976 to 1984, the frequency of observed deer (based on the interview survey) increased. Since 2005, population estimations by light census started and about 20-50 individuals/km2 were confirmed from 2005-2008. I hear the census is continuing, but no reports have been released recently.

In Material and Methods, what are the soil types in different forests?

[Res] The soil group of the two study regions is andosole according to the classification system of the IUSS Working Group World Reference Base (2014). We added the explanation about it (L100-102)

In Results, it is suggested to divide Environmental properties into two paragraph, e.g. canopy openness, and soil physical and chemical properties.

[Res] We divided the paragraph accordingly (L247-274)

In Line 276, 34 phyla were confirmed in two kingdoms, however, there are only top five phyla and 14 phyla in fungal community, what are the rest phyla?

[Res] In this paragraph, we showed the results of the prokaryote community, not the fungal community.

The top five phyla belonged to prokaryote phyla. We are sorry that the word “kingdoms” would cause a misunderstanding, and we changed the word to “domains”. For the response to this comment and the next comment, we changed the sentence from

“The amplicon sequencing of the 16S V3–V4 regions confirmed 34 phyla in two kingdoms”

to

“The amplicon sequencing of the prokaryotic 16S rRNA V3–V4 regions confirmed 34 phyla in two domains (archaea and bacteria)” (L277-278).

In Line 277, are the top five phyla belong to prokaryote? How many phyla in prokaryote?

[Res] Yes, the five phyla belong to prokaryotes. 34 phyla were confirmed in prokaryote (L278, S2 Table). 

In Discussion, there is no contents about canopy openness in the first paragraph, only soil properties. It is related to soil temperature, soil water and microbiome activity.

[Res] We further reviewed the previous studies about Ericaceae dominant effects on the environment. And we added the discussion for light quantity (PAR) and quality (R:FR) changes (L405-414).

 As you kindly commented, other environmental variables, such as soil water contents and soil temperatures, can be changed by the understory vegetation (e.g., grass, dwarf bamboo). However, for the soil water contents, Ericaceae dominance does not necessarily change the soil water content (Clinton et al. 2003).

 For the soil temperatures, we could not find the previous studies on the soil temperature changes by the Ericaceae dominances. To focus on the environmental changes caused by the Ericaceae dominance, we discussed the light environmental changes in this discussion (L405-414).

 For microbiome activities, microbial activities have been sometimes measured as the soil respirations or enzyme activities, and the previous study showed that the presence of Ericaceae plant affected enzyme activities and soil C and N cycles (Osburn et al. 2018). Although our dataset showed that non significant relationships between Pieris dominances and the prokaryote community and soil C and N, we would like to introduce the previous study as a topic for the next study (L545-551) 

Line 431-432, Multiple types of mycorrhizae fungi have been found in the roots of other ericaceous plants using molecular identification (e.g., Matsuda et al. 2012; Baba et al. 2016). Invalid citation. What are the main types of mycorrhizae fungi in these studies?

[Res] 

For Matsuda et al. 2012, they identified the root-associated fungi of Pyrola japonica (Ericaceae), and they found Russula sp. which is the ectomycorrhizae fungi and dominated in the plant’s root. The main type of mycorrhizae fungi of this species is ECM. However, they also find the Hymenoscyphus sp. (Helotiales, which contains both ECM and ErM fungi) and Glomus sp. which is the arbuscular mycorrhizae fungi in roots with a few rates.

For Baba et al. 2016, they found ErM, such as Helotiales sp. and Oidiodendron sp. in the roots of Vaccinum oldhamii (Ericaceae) and reported that the main type of mycorrhizae fungi of the species is ErM. However, they also found the Russula sp. (ECM fungi) and Rhizophagus (AM fungi) from a few of the roots.

These studies suggest that Ericaceae have mutualistic relationships with multiple mycorrhizae fungi (ErM, ECM, and AM).

We are sorry we could not understand how invalid citations for these papers are, and we do not have confidence whether our response is what you would like to receive. We added the scientific names of the plants and mycorrhizae fungal types in parenthesis and kept the sentences as were (L475-476).

In Conclusions and suggested future studies on Pieris-dominant forests, it is suggested to move future studies into Discussion. Conclusion need to be focused on soil change.

[Res] We moved the paragraphs about future studies into Discussion. We reconstructed the Conclusion sections focusing on the forest environment and soil microbial community changes by Pieris dominances.

Line 488-489, A drawback of Pieris-dominant areas is that they inhibit the establishment of fore regeneration. Which table or figure support this conclusion? There is no direct result.

[Res] We did not show the table or figure supporting the issue. We deleted the paragraph which claimed the drawbacks of Pieris-dominant areas from the conclusion. 

The issues of the drawbacks have been discussed partly in the discussion section, and we rearranged the discussion section including future studies.

Line 498-499, A potential advantage of Pieris is that it may play an important role in preventing soil erosion, in terms of soil bulk density and subsequent prokaryotic communities. Please focus on your findings. You did not measure soil erosion.

[Res] Thank you again for the comments on this issue. We deleted the paragraph which claimed the potential advantage of Pieris from the conclusion. The issue has been discussed partly in the discussion, so we rearranged the discussion section.

We hope our responses could meet your criteria for evaluation and revision.

Sincerely,

Yuji Tokumoto

---

## [Decision Letter · Decision Letter 1]

1 Dec 2023

PONE-D-23-27922R1Effects of Pieris japonica (Ericaceae) dominance on cool temperate forest altered-understory environments and soil microbiomes in Southern JapanPLOS ONE

Dear Dr. Tokumoto,

Thank you for submitting your manuscript to PLOS ONE. After careful consideration, we feel that it has merit but does not fully meet PLOS ONE’s publication criteria as it currently stands. Therefore, we invite you to submit a revised version of the manuscript that addresses the points raised during the review process.

**ACADEMIC EDITOR: **The revised version has been improved a lot.  But the manuscript still has some problems as suggested by the reviewer.

We look forward to receiving your revised manuscript.

Kind regards,

Jian Liu

Academic Editor

PLOS ONE

Journal Requirements:

Additional Editor Comments:

The revised version has been improved a lot. But the manuscript still has some problems as suggested by the reviewer. The authors should respond to the comments of the reviewers one by one and revise the manuscript accordingly.

Reviewers' comments:

Reviewer's Responses to Questions

**Comments to the Author**

1. If the authors have adequately addressed your comments raised in a previous round of review and you feel that this manuscript is now acceptable for publication, you may indicate that here to bypass the “Comments to the Author” section, enter your conflict of interest statement in the “Confidential to Editor” section, and submit your "Accept" recommendation.

Reviewer #1: All comments have been addressed

Reviewer #2: All comments have been addressed

2. Is the manuscript technically sound, and do the data support the conclusions?

Reviewer #1: Yes

Reviewer #2: Yes

3. Has the statistical analysis been performed appropriately and rigorously? 

Reviewer #1: Yes

Reviewer #2: Yes

4. Have the authors made all data underlying the findings in their manuscript fully available?

Reviewer #1: Yes

Reviewer #2: Yes

5. Is the manuscript presented in an intelligible fashion and written in standard English?

Reviewer #1: Yes

Reviewer #2: Yes

6. Review Comments to the Author

Reviewer #1: (No Response)

Reviewer #2: In this version, the authors answered most questions from the editor and two reviewers and improved the quality of manuscript. There are only some minor questions need to be correct before publication.

In Environmental properties, if an index is significant between different sites, it is suggested to declare in which site it is higher or lower.

Line 545-547, A previous study has compared the extracellular enzyme activities between sites with or without Ericaceae removal, and the presence of Ericaceae affects the microbial activities and soil C and N cycles [56]. Invalid citation. What is the main findings of this study? Did the presence of Ericaceae increase or decrease microbial activities, soil C and N cycles? Please check other citations, especially in discussion.

Line 252-254, the reason for canopy openness change should be presented in discussion, not in results.

There are some minor errors in format.

Table 2, the decimal of all data should be 0.01.

Line 150, Table 2, Line 235, Line 264, Line 312, Line 316, Line 319, Line 320, Line 321, Line 334, Line 347, Line 531, Line 535, Line 543, SOMcontent should be SOM content, a space symbol was missing.

Table 3, the decimal of all data should be 0.001.

Table 2, Table 3, Table 4, P-values should be p-values.

The first subtitle of discussion should be more concise. For example, Differences between understory environments according to Pieris dominance for different canopy trees.

Line 471, Pieris, the total word should be italic.

Line 588, 13 should not be bold.

Line 600, Line 606, Line 611, the article title should not be ended with a comma “,”.

Line 713, Line 745, last full stop “.” should be deleted.

7. PLOS authors have the option to publish the peer review history of their article (what does this mean?). If published, this will include your full peer review and any attached files.

Reviewer #1: No

Reviewer #2: No

---

## [Author Response · Author response to Decision Letter 1]

15 Dec 2023

Response to the academic editor and reviewers’ comments on Ms. No. PONE-S-23-32909 Effects of Pieris japonica (Ericaceae) dominance on cool temperate forest altered-understory environments and soil microbiomes in Southern Japan

Response to the academic editor’s comments

Dear Professor, Dr. Jian Liu, Academic Editor of PLOS ONE,

Thank you very much for your review of our manuscripts.

We would like to reply to your comments below. 

Journal Requirements:

[Res] We reviewed the reference list to complete the correction. We changed the list as below.

- We enclosed the Issue number of Malik AU 2003 in parentheses.

- We made the mistake of citing two articles: Chamber et al. 2008 [9] and Walker et al. 1999 [46] while writing. We would like to replace Chamber et al. with Walker et al. and vice versa. We apologize for the mistake.

- The same reference was cited twice (12 and 29). After deleting 29, we corrected the number of citations after 29. 

- The articles from 38 to 40 were changed in the order they appeared.

- Katayama et al. [41] have been published in the journal and we added issue and page numbers.

Additional Editor Comments:

The revised version has been improved a lot. But the manuscript still has some problems as suggested by the reviewer. The authors should respond to the comments of the reviewers one by one and revise the manuscript accordingly. 

[Res] Thank you very much for your comments. We amended the manuscript files based in the reviewer’s comments. We would like you to confirm the documents again and would be grateful if the revised manuscript would meet the standard for the publication of PLOS ONE.

Sincerely,

Yuji Tokumoto, on behalf of the authors

 

Response to the reviewers’ comments

Dear Reviewer #2:

Thank you very much for the review and positive and constructive comments again. We would like to reply to the comments below.

Reviewer #2: In this version, the authors answered most questions from the editor and two reviewers and improved the quality of manuscript. There are only some minor questions need to be correct before publication.

In Environmental properties, if an index is significant between different sites, it is suggested to declare in which site it is higher or lower.

[Res] We modified the sentences that mentioned total litter weight, humus and Pieris leaf weights, soil bulk density and SOM content (second paragraph of section Environmental properties in Results). 

Line 545-547, A previous study has compared the extracellular enzyme activities between sites with or without Ericaceae removal, and the presence of Ericaceae affects the microbial activities and soil C and N cycles [56]. Invalid citation. What is the main findings of this study? Did the presence of Ericaceae increase or decrease microbial activities, soil C and N cycles? Please check other citations, especially in discussion.

[Res] We modified the sentences that mentioned [55] (the previous number of the reference is 56) (L549-552). We checked the other citations throughout the text and found similar descriptions in (L55-57 and L523-524), so we also modified those sentences to introduce the findings of previous studies precisely.

Line 252-254, the reason for canopy openness change should be presented in discussion, not in results.

[Res] We moved the reason for canopy openness change to the first paragraph of the discussion. We have discussed the unfluctuating light conditions under the Pieris populations in L405-410.

There are some minor errors in format.

Table 2, the decimal of all data should be 0.01.

[Res] We amended it accordingly.

Line 150, Table 2, Line 235, Line 264, Line 312, Line 316, Line 319, Line 320, Line 321, Line 334, Line 347, Line 531, Line 535, Line 543, SOMcontent should be SOM content, a space symbol was missing.

[Res] We amended it accordingly.

Table 3, the decimal of all data should be 0.001.

[Res] We amended it accordingly.

Table 2, Table 3, Table 4, P-values should be p-values.

[Res] We amended it accordingly.

The first subtitle of discussion should be more concise. For example, Differences between understory environments according to Pieris dominance for different canopy trees.

[Res] We modified the subtitle to be concise as you kindly suggested. 

Line 471, Pieris, the total word should be italic.

[Res] We amended it accordingly.

Line 588, 13 should not be bold.

[Res] We amended it accordingly.

Line 600, Line 606, Line 611, the article title should not be ended with a comma “,”.

[Res] We amended it accordingly. We found the same mistake in [3] and amended it.

Line 713, Line 745, last full stop “.” should be deleted.

[Res] We amended it accordingly.

Thank you for your review in details.

We hope our responses could meet your criterion for evaluation and revision.

Sincerely,

Yuji Tokumoto

---

## [Editor Report · Decision Letter 2]

18 Dec 2023

Effects of Pieris japonica (Ericaceae) dominance on cool temperate forest altered-understory environments and soil microbiomes in Southern Japan

PONE-D-23-27922R2

Dear Dr. Tokumoto,

We’re pleased to inform you that your manuscript has been judged scientifically suitable for publication and will be formally accepted for publication once it meets all outstanding technical requirements.

Kind regards,

Jian Liu

Academic Editor

PLOS ONE

Additional Editor Comments (optional):

All the comments have been addressed.
---

## [Editor Report · Acceptance letter]

4 Jan 2024

PONE-D-23-27922R2 

PLOS ONE

Dear Dr. Tokumoto, 

I'm pleased to inform you that your manuscript has been deemed suitable for publication in PLOS ONE. Congratulations! Your manuscript is now being handed over to our production team.

Kind regards, 

on behalf of

Dr. Jian Liu 

Academic Editor

PLOS ONE